# High Piezoelectric Output Voltage from Blue Fluorescent *N*,*N*-Dimethyl-4-nitroaniline Nano Crystals in Poly-L-Lactic Acid Electrospun Fibers

**DOI:** 10.3390/ma15227958

**Published:** 2022-11-10

**Authors:** Rosa M. F. Baptista, Bruna Silva, João Oliveira, Vahideh B. Isfahani, Bernardo Almeida, Mário R. Pereira, Nuno Cerca, Cidália Castro, Pedro V. Rodrigues, Ana Machado, Michael Belsley, Etelvina de Matos Gomes

**Affiliations:** 1Centre of Physics of Minho and Porto Universities (CF-UM-UP), University of Minho, Campus de Gualtar, 4710-057 Braga, Portugal; 2Centre of Biological Engineering, University of Minho, Campus de Gualtar, 4710-057 Braga, Portugal; 3Institute for Polymers and Composites, University of Minho, Campus de Azurém, 4800-058 Guimarães, Portugal

**Keywords:** electrospinning, microfibers, piezoelectric crystals, nitroanilines, fluorescence, functional organic materials

## Abstract

*N*,*N*-dimethyl-4-nitroaniline is a piezoelectric organic superplastic and superelastic charge transfer molecular crystal that crystallizes in an acentric structure. Organic mechanical flexible crystals are of great importance as they stand between soft matter and inorganic crystals. Highly aligned poly-l-lactic acid polymer microfibers with embedded *N*,*N*-dimethyl-4-nitroaniline nanocrystals are fabricated using the electrospinning technique, and their piezoelectric and optical properties are explored as hybrid systems. The composite fibers display an extraordinarily high piezoelectric output response, where for a small stress of 5.0 × 10^3^ Nm^−2^, an effective piezoelectric voltage coefficient of *g*_eff_ = 4.1 VmN^−1^ is obtained, which is one of the highest among piezoelectric polymers and organic lead perovskites. Mechanically, they exhibit an average increase of 67% in the Young modulus compared to polymer microfibers alone, reaching 55 MPa, while the tensile strength reaches 2.8 MPa. Furthermore, the fibers show solid-state blue fluorescence, important for emission applications, with a long lifetime decay (147 ns) lifetime decay. The present results show that nanocrystals from small organic molecules with luminescent, elastic and piezoelectric properties form a mechanically strong hybrid functional 2-dimensional array, promising for applications in energy harvesting through the piezoelectric effect and as solid-state blue emitters.

## 1. Introduction

Organic acentric crystals are excellent candidates for applications as piezoelectric, nonlinear and electro-optical materials. Those containing nitroaniline derivative molecules are the simplest and exhibit the highest piezoelectric and nonlinear optical properties. The push–pull conjugated charge-transfer, D^+^-π-A^−^, molecular systems formed by intramolecular electron donor (D) acceptor (A) groups can form supramolecular assemblies due to molecular dipole–dipole interactions, mediated by hydrogen bonds and other noncovalent interactions. These molecules have very high polarizabilities and hyperpolarizabilities and are among the best candidates for a wide range of piezoelectric and photonic applications [1,2,3,4].

The archetype charge transfer (CT) nitroaniline isomeric molecules are para-nitroaniline or 4-nitroaniline (4 NA or pNA, C_6_H_6_N_2_O_2_) and meta-nitroaniline or 3-nitroanilie (3 NA or mNA, C_6_H_6_N_2_O_2_) with dipole moments of µ = 8.2 D and µ = 6.9 D, respectively [5]. While pNA crystallizes in the centrosymmetric point group 2/*m* [6] due to a pairwise molecular antiparallel alignment in the crystal unit cell, mNA crystallizes in the acentric point group mm2. Piezoelectricity is therefore forbidden by symmetry in pNA, while mNA has one of the largest bulk piezoelectric coefficients measured on a molecular crystal, *d*_31_ = 31 × 10^−12^ NC^−1^. It is also a strong electro-optical and nonlinear optical organic crystal [7,8].

Adding to the molecules π-electron-donating groups, such as OCH_3_, NH_2_ and N(CH_3_)_2_, leads to an increased asymmetric charge distribution resulting in higher dipole moments, polarizability and hyperpolarizability [9]. An example is 2-methyl-4-nitroaniline (MNA, C_7_H_8_N_2_O_2_) with a large molecular dipole moment equal to µ = 8.1 D and hyperpolarizability *β* = 71 × 10^−40^ m^4^V^−1^ at the wavelength of maximum absorption of 361 nm. It crystallizes in the acentric point group m and displays in the solid state a crystal dipole moment of µ = 19.5 D [10], originating a large piezoelectric coefficient, *d*_31_ = 13.8 × 10^−12^ NC^−1^ [11]. MNA is a prototype nonlinear optical and electro-optic crystal [12,13,14]. 

Another engineered pNA derivative molecule, although much less attention has been given to it, is *N*,*N*-dimethyl-4-nitroaniline (NNDM4NA, C_8_H_10_O_2_N_2_), with each hydrogen atom in the NH_2_ group replaced by a CH_3_ group, and the dimethyl and nitro groups at the *para* position on the benzene ring, which crystallizes in the polar point group 2 (Space group P2_1_) [15]. The molecule also has a high molecular dipole moment, µ = 7.95 D, similar to MNA [16], Figure 1a–d. Very importantly, NNDM4NA crystals exhibit superplastic and superelastic properties through slip and twinning deformations dependent on the applied force direction: at room temperature, superplasticity occurs when strain is applied along <100> and is accompanied by superelastic behavior along <201>, keeping single crystallinity during deformation [17]. Superplasticity, the ability to exhibit exceptionally large elongations during tensile deformation, is a well-known property displayed by metals, alloys and polymers [18,19]. However, those are rare properties of organic crystals, which are presently an important research area, as plasticity and flexibility are of most importance for applications such as mechanical actuators, bioelectronics and photonics [20,21,22,23,24].

Electrospinning is an effective nanofabrication technique in which an electrohydrodynamic thrust created by an intense electrostatic field allows the production of continuous micro/nanofibers from the extrusion of a charged polymer fluid. The polymer fibers have a high surface-to-volume ratio, are lightweight, and have an interconnected porous structure. The experimental accessibility and versatility of the technique allow the preparation of nanomaterials from a remarkably broad class of macromolecular-based systems [25,26,27]. Therefore, the functional properties of electrospun fibers are determined by the chemical nature of the solutions of electrospinning materials, the different processing conditions as well as the crystallinity of the polymer and its molecular orientation within micro/nanofibers [28,29]. The technique has been used as a method for producing aligned polymer-doped micro/nanofibers with a high degree of guest molecular polar orientation within the polymer. A large variety of organic molecules, from small nitroaniline derivatives to supramolecular dipeptide molecules, with enhanced piezoelectric and nonlinear optical second harmonic generation, have been embedded in electrospun fibers [28,30,31,32,33,34,35,36,37].

Studies of 3NA nanocrystals embedded in poly-ε-caprolactone (PCL) polymer nanofibers showed an enhanced SHG response and piezoelectric output voltage as a mechanical to electrical energy converter, with an instantaneous density power of magnitude 122 nWcm^−2^ similar to that displayed by poly(vinylidene fluoride) and poly(γ-methyl L-glutamate) composite fibers [34].

PLLA is a biodegradable and biocompatible chiral polymer containing two 10/3 helical chains along the crystallographic axis c of an orthorhombic base-centered unit cell and a huge optical rotatory power (9000° mm^−1^) [38] (Figure 1e). Oriented films of PLLA show a piezoelectric bulk shear coefficient *d*_14_ = 10 pCN^−1^ [39].

In this work, NNDM4NA nanocrystals are embedded in aligned PLLA microfibers (NNDM4NA@PLLA), fabricated using the electrospinning technique, forming a hybrid functional material. We reported its extraordinarily high piezoelectric output response, where for an applied periodical force of 1.5 N (stress 5.0 × 10^3^ Nm^−2^), an effective piezoelectric voltage coefficient of *g*_eff_ = 4.1 VmN^−1^ was obtained, five times higher than that of a two-dimensional hybrid ferroelectric perovskite of (4-aminotetrahydropyran)_2_ lead bromide (ATHP)_2_PbBr_4_), which exhibited an already large value of *g*_33_ = 0.67 VmN^−1^ [40]. Very high value resulted from the superplastic and piezoelectric properties of NNDM4NA nanocrystals together with the piezoelectric properties of PLLA polymer. Furthermore, both in solution and when embedded in electrospun PLLA fibers, NNDM4NA@PLLA displays strong blue fluorescence with a 147-ns-long lifetime decay. In this paper, we explore the optical and piezoelectric properties of *N*,*N*-dimethyl-4-nitroaniline nanocrystals as hybrid functional systems.

## 2. Materials and Methods

### 2.1. Solution Preparation

*N*,*N*-dimethyl-4-nitroaniline (NNDM4NA) was purchased from Sigma-Aldrich (St. Louis, MI, USA) and used as received. Poly (L-lactic acid) (PLLA, Mw 217-225000) was purchased from Polysciences. All solvents were purchased from Sigma-Aldrich and used as received. The precursor electrospinning solution was prepared by dissolving in 4 mL of dichloromethane (DCM) 0.5 g of PLLA with vigorous stirring (700 rpm) at 40 °C for 1 h (IKA RCT basic, Staufen, Germany). After complete dissolution, 0,17 g of NNDM4NA previously dissolved in 1 mL of dimethylformamide (DMF) was added to the polymer solution. The resulting solution with a concentration of 10% by weight was stirred for several hours under ambient conditions prior to the electrospinning process.

### 2.2. Electrospinning of Microfibers

NNDM4NA@PLLA composite microfibers (1:3) were electrospun using a conventional electrospinning apparatus described in detail before [30]. The previously prepared clear and homogenous solution was loaded into a plastic syringe with a blunt-ended needle with a 0.5 mm inner diameter. The syringe was located on a syringe pump and connected to the anode of a high-voltage power supply CZE2000 (Spellman, Bochum, Germany). Electrospinning was performed at room temperature, and various parameters were adjusted to obtain bead-free fibers and stable spinning conditions, namely, the solution feeding flow rate, the electric potential difference, and the needle tip-collector distance. An electric potential difference of 18 kV was established between the syringe needle and a grounded rotatory collector. The needle-collector distance was 12 cm, and the flow rate was 0.19 mL/h. The aligned fibers were collected on a rotatory mandrel rotating at a rate of 300 rpm coated with high-purity aluminum foil. The fibers collected on a rapidly rotating mandrel were oriented along the rotating direction of the mandrel due to the mechanical stretching forces inherent to the fabrication process.

### 2.3. Characterization Techniques

The crystallinity and crystallographic orientation of *N*,*N*-dimethyl-4-nitroaniline were studied by XRD. The diffraction pattern using θ–2θ scans was recorded between 5° and 40° on a Philips PW-1710 X-ray diffractometer with Cu-K_α_ radiation of wavelength 1.5406 Å. The lattice planes parallel to the substrate surface were determined from the reciprocal lattice vector of modulus (2/λ)sin θ, with λ the radiation wavelength and θ the Bragg angle of diffraction. Morphology, diameter distribution, and fiber thickness were accessed through a Nova NanoSEM scanning electron microscope operated at an accelerating voltage of 10 kV. NNDM4NA@PLLA microfibers were deposited on a silica surface and previously covered with a thin film (10 nm thickness) of Au-Pd (80–20 weight%) using a high-resolution sputter cover, 208 HR Cressington Company, coupled to an MTM-20 Cressigton high-resolution thickness controller. The diameter range of the microfibers produced was measured through SEM images using ImageJ 1.51 n image analysis software (NIH, https://imagej.nih.gov/ij/, 23 June 2022). The average diameter and diameter distribution were determined by measuring 79 random microfibers from the SEM images and fitting the fiber diameter distributions to a lognormal function. The thickness of the fiber array was measured using a Mitutoyo IP 65 micrometer.

Elastic modulus, stress at yield (at 0.2% offset), tensile strength and strain at break (at 60% of tensile strength) were measured using a universal tensile testing machine Zwick/Roell Z005, following the ASTM D882-02 standard. Several 10 × 30 mm samples, with a gauge length of 16 mm, were tested along the oriented fiber direction with a cross-head velocity of 25 mm/min.

Optical absorption (OA) spectra were measured on a Shimadzu UV/2501PC spectrophotometer (Shimadzu, Duisburg, Germany) in the wavelength range of 250–480 nm. Photoluminescence (PL) spectrum, measured at an excitation wavelength corresponding to the maximum absorption wavelength, was collected using a FluoroMax-4 spectrophotometer (Horiba Jovin Yvon, Madrid, Spain) in the wavelength range 400–750 nm at room temperature. For these measurements, a 10^−4^ M ethanol solution of NNDM4NA was prepared and placed in quartz cuvettes with a 10^−2^ m path length. The input and output slits were fixed at 5 nm. The fiber luminescence was also observed with an OlympusTM FluoView FV1000 (Olympus, Tokyo, Japan) confocal scanning laser microscope, using a 40× objective and excitation wavelength of 405 nm.

The dielectric properties were characterized by impedance spectroscopy at temperatures from 15 °C to 100 °C and in the frequency range 10 Hz–3 MHz. In order to perform the measurements, we connected the nanofiber array samples, forming a parallel plate capacitor included in an LCR network. The electrical contacts on the top and bottom surfaces of the mats were 10 mm in diameter and were made with air-cured silver paste. The complex permittivity ε = ε′ − iε″, with ε′ and ε′′ the real and imaginary parts, respectively, calculated from the measured capacitance, C = ε′ε_0_ (A/*d*) and loss tan *δ* = ε′′/ε′ with A the contact area, ε_0_ the vacuum permittivity and *d* the thickness of the array. A Wayne Kerr 6440 A precision component analyzer connected to a dedicated computer and software was used to acquire the data [41]. Shielded test leads were used to avoid parasitic impedances from connecting cables. Temperature-dependent measurements were performed with a rate ramp of 2 °C/min, using a Polymer Labs PL706 PID controller and furnace.

Piezoelectric output voltage and current were measured across a 100 MΩ load resistance connected to a low-pass filter followed by a low-noise preamplifier (Research systems SR560, Stanford Research Systems, Stanford, CA, USA) before being registered with a digital storage oscilloscope (Agilent Technologies DS0-X-3012A, Waldbronn, Germany). The fiber array sample with a 20 × 20 mm^2^ area (600 µm thickness) was subjected to applied periodic mechanical forces imposed by a vibration generator (Frederiksen SF2185) with a frequency of 3 Hz determined by a signal generator (Hewlett Packard 33120A). The forces applied were previously calibrated using a force-sensing resistor (FSR402, Interlink Electronics Sensor Technology, Graefelfing, Germany). The fibers were directly deposited, during the electrospinning process, on high-purity aluminum foil, which served as electrodes. The samples were fixed on a stage, and the forces were applied uniformly and perpendicularly over the surface area of each sample. A schematic setup is shown in Figure 6c).

## 3. Results and Discussion

### 3.1. Fiber Morphology and Crystallinity

The microfibers are colored yellow in daylight and form a two-dimensional array of highly aligned fibers with a mean diameter of 1.38 µm (Figure 1a–c). The surface of each fiber is smooth (Figure 1d), with no beads or crystals grown on it, and all NNDM4NA crystals are in the fiber interior and evenly spread inside the fibers, as the confocal microscopy images show in Figure 2b. These mutually aligned fibers are the result of both the chosen electrospinning processing parameters combined with the mechanical stretching forces present during collection on a rotatory drum, as previously reported for MNA_PLLA fibers [30].

### 3.2. Optical Absorption and Luminescence

An NNDM4NA ethanol solution presents an intense absorption band in the range of 300–450 nm, with a maximum wavelength of 386 nm arising from CT π-π* transitions [42,43]. Excitation at the maximum absorption wavelength originates in the PL spectrum, an intense emission band with a maximum of 460 nm, in the blue region, with a full width at half-maximum of 67 nm (see Figure 2) and a Stokes shift of 74 nm. The PL spectrum of an NNDM4NA@PLLA fiber array shows a maximum emission wavelength at 510 nm, with a full width at a half-maximum of 70 nm, exhibiting a red shift of 124 nm (an additional 50 nm relative to that in solution) (Table 1). This high bathochromic shift towards the long wavelength of CT transition indicates a more extended conjugation, which may be attributed to the stretched molecular conformations along the longitudinal fiber axis as a result of extensional flow during the electrospinning process. FTIR results are in accordance with these conclusions, as shifts from some bands were observed for the fiber array.

The emission from a nanofiber array was further studied by fluorescence confocal microscopy with excitation at 405 nm. The embedded NNDM4NA nanocrystals displayed a bright blue light that revealed the uniformity of the emission along the fibers (Figure 2b). The strong blue emission of NNDM4NA@PLLA fibers must have been favored by the alignment and stretching of the fibers imposed by the rotatory drum collector during the electrospinning process. It was reported that the collection of electrospun microfibers on a rotatory drum originates both aligned and well-stretched fibers with increased photoluminescence [44], which, together with a more effective molecular packing at the nanoscale, contributes to the Stokes shift of the observed emission [33]. The emission at blue wavelengths of organic fluorophores in their condensed phase is of great scientific importance in materials science, towards their application as optical sensors and in luminescent and light-emitting devices [45,46,47,48]. Only a few compounds have a significant increase in light emission in their solid state, which overcomes the problem of aggregation quenching. Importantly, NNDM4NA nanocrystals embedded in PLLA electrospun fibers are, therefore, one of them.

The fluorescence lifetime decay studied in detail, using single photon counting equipment, showed that there is a main component of fluorescence (Appendix A) of the NNDM4NA@PLLA fiber array, which accounts for 92% of all fluorescence emitted with a lifetime decay of 147 ns. The fluorescence also has a small contribution (8% total) from two other components, both with very short decay times (less than 5 ns), as indicated in Appendix A.

### 3.3. Attenuated Total Reflectance Fourier Transform Infrared Spectroscopy (FTIR-ATR)

The Fourier transformed infrared (FTIR) analysis of electrospun fibers and NNDM4NA crystalline powder was recorded on a Jasco 4100 FTIR spectrometer in transmittance mode, in the range of 4500 to 600 cm^−1^, by averaging 32 scans and using a resolution of 8 cm^−1^. FTIR data were treated with OriginPro 2017 SR2 software (OriginLab Corporation, Northampton, MA, USA). The incorporation of NNDM4NA in PPLA polymeric fibers was confirmed spectroscopically by the collected data, as indicated in Figure 3. NNDM4NA exhibits the characteristics of conjugated double bonds with typical bands within the ranges 1660–1480 cm^−1^, which is consistent with a high degree of push–pull structural character (donor–acceptor). NNDM4NA, as a tertiary amine, has C-N stretching bands from 1250 cm^−1^ to 1020 cm^−1^. In the NNDM4NA spectrum, two bands appear at 1060 cm^−1^ and 1095 cm^−1^, with weak to medium intensity, due to the lack of polarity of the C-N bond. These bands overlap with the more intense PLLA bands in the NNDM4NA@PLLA spectrum. For aromatic nitro compounds, the asymmetric and symmetric stretching vibration of the NO_2_ group is assigned to an intense band in the region around 1590–1600 cm^−1^ and 1380–1390 cm^−1^, which is due to the asymmetric and symmetric stretching vibration of the group [49,50,51,52,53]. The symmetric stretching vibration at 1304 cm^−1^ of NNDM4NA is shifted to 1315 cm^−1^ in NNDM4NA@PLLA [54,55]. The bands in the 1123 cm^−1^ and 1177 cm^−1^ regions are assigned to bending vibrations of the C-H and C-C. Weak bands at 1432 cm^−1^ and 1462 cm^−1^ in NNDM4NA are not present in NNDM4NA@PLLA due to overlapping with the intense PLLA band at 1452 cm^−1^. The band at 1502 cm^−1^ in NNDM4NA, also associated with NO_2_ asymmetric stretching vibration, is shifted to 1529 cm^−1^ in NNDM4NA@PLLA. The weak 1540 cm^−1^ band in NNDM4NA is hidden by the polymer matrix in NNDM4NA@PLLA. In the NNDM4NA spectrum, the intense band at 1657 cm^−1^, which is mainly due to the in-plane bending modes of N(CH_3_)_2_, is absent in the NNDM4NA@PLLA spectrum. The reason must be that the nanocrystals are intercalated in the polymer chains, and those groups are constrained inside the matrix.

### 3.4. Mechanical Performance

The results of the tensile characterization are shown in Figure 4. NNDM4NA@PLLA fibers exhibit good mechanical performance: there is an average increase of 67% on the Young modulus reaching 55 MPa, while the tensile strength reaches 2.8 MPa when compared to PLLA fibers. Both the stress at yield and the tensile strength has a positive variation of 50%. This significant increase in tensile strength and Young modulus of doped electrospun fibers indicates that hybrid NNDM4NA@PLLA arrays are stiffer and mechanically stronger than neat PLLA fibers.

This increase must be related to the superelastic properties reported for NNDM4NA crystals, where for a small tensile strain (inferior to 100%), there is an increase in tensile stress [17]. There is a decrease of 53% of the strain at break for the doped fibers indicating a decrease in their plasticity. This may be explained by the presence of NNDM4NA nanocrystals inside the polymer matrix that act as a discontinuous phase inhibiting the ability of PLLA chains to flow between themselves, thus limiting the creep of molecular chains accompanied by an increase in the rigidity of the polymer chain [56].

### 3.5. Dielectric Permittivity

Figure 5a shows the temperature dependence of the real part of the dielectric permittivity (ε′) for the composite, NNDM4NA@PLLA fibers, and the respective measurements in pure PLLA microfibers. Both curves follow similar trends: up to 45 °C, there is no variation in the dielectric constant with temperature, afterward increasing steeply with two-step-like anomalies, one at ~55 °C and the other at ~65 °C, which correspond to the glass transition temperature of PLLA fibers [57,58].

The pure PLLA polymer glass transition temperature has been reported to occur in the Tg range, bulk = 55–65 °C [59]. There is, for NNDM4NA@PLLA fibers, an increase the dielectric constant with the temperature reaching 7 for frequencies between 100 Hz and 1 MHz relative to pure PLLA fibers. At room temperature and for all frequencies, it is around 3.5 for NNDM4NA@PLLA fibers, while for NNDM4NA pellet, it is around 6, Figure 5a,b, respectively. Electrospun NNDM4NA fibers have a dielectric constant significantly higher (approximately 75%) than pure PLLA fibers, Figure 5a, which results from the polycrystalline contribution of NNDM4NA, which confirms its successful incorporation within the polymer matrix.

### 3.6. Microfibers Piezoelectric Voltage

The piezoelectric effect is a phenomenon that arises when an applied uniform stress generates electric polarization inside a dielectric material, where an interconversion between mechanical and electrical stimulus is present. For a crystalline solid to display the phenomenon, it must have a crystal structure without inversion symmetry. NNDM4NA crystallizes in the polar point group 2, which is acentric and consequently allows piezoelectricity. The tensor relationship between the polarization *P*_i_ and the stress σ_j_ tensors j is given by *d*_ij_, the piezoelectric modulus according to the relation (written in the matrix notation) *P*_i_ = *d*_ij_ σ_j_ [60]. Inside the electrospun fibers, there is no preferential crystallographic orientation, and the periodic forces were applied perpendicular to the fiber array and measured along the same direction; see Figure 6c. Therefore, there is an overall polarization such that the piezoelectric modulus tensor is an effective coefficient, *P* (NNDM4NA@PLLA) = *d*_eff_ σ. The applied stress ranged between 1.3 × 10^3^ Nm^−2^ and 11.3 × 10^3^ Nm^−2^.

**Figure 6 materials-15-07958-f006:**
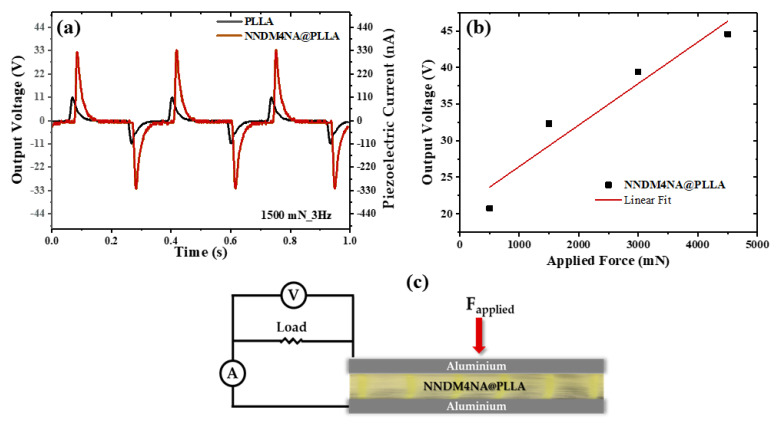
(**a**,**b**) Output voltage (V) and current (nA) measured through a load resistance of 100 MΩ on the electrospun fiber array of PLLA and PLLA doped with NNDM4NA nanocrystals, (**c**) and schematic piezoelectric setup for the NNDM4NA@PLLA fiber array.

Extraordinarily, for NNDM4NA@PLLA fiber mats, a 1.5 N applied periodical force originates, respectively, a 33 V and 330 nA maximum instantaneous output piezoelectric voltage and current, as shown in Figure 6a. Here, the two opposite peaks correspond respectively, to the exerted and released force. A plot of the output voltage as a function of several applied periodic forces shows a linear increase in the response with the magnitude of the force as expected, Figure 6b. We should note that there is a contribution from the PLLA polymer matrix of 100 nA for the same applied force, as PLLA is itself a piezoelectric polymer. Under similar conditions, electrospun nanofiber mats fabricated with 3 NA embedded into poly-ε-caprolactone (PCL) polymer, 3NA@PCL, originated an output voltage and current of 5 V and 50 nA, respectively [34]. Remarkably, the output voltage of NNDM4NA@PLLA is more than six times higher.

We can calculate the magnitude of *d*_eff_ from the integration of the induced piezoelectric current over a time period of 1 ms, Q = I dt. From our measurements, Figure 6a, we obtained Q = 330 pC for NNDM4NA@PLLA fiber mats and Q = 220 pC considering only the contribution of NNDM4NA nanocrystals. The induced charge is related to the applied force by the equation Q = *d*_eff_ F. Therefore, an effective piezoelectric coefficient equal to *d*_eff_ = 220 pCN^−1^ for an NNDM4NA@PLLA fiber array and *d*_eff_ = 147 pCN^−1^ for nanocrystalline NNDM4NA, disregarding the polymer contribution. These values are extraordinarily high for an organic crystal which compares well with those of organic–inorganic ferroelectric perovskites trimethylchloromethyl ammonium trichloromanganese (TMCM)MnCl_3_ and trimethylchloromethyl ammonium trichlorocadmium (TMCM) CdCl_3_, respectively, *d*_33_ = 230 pCN^−1^ and *d*_33_ = 185 pCN^−1^, and barium titanate (BaTiO_3_) with *d*_33_ = 190 pCN^−1^ [61,62].

In addition to the magnitude of the piezoelectric coefficient, another relevant quantity is the piezoelectric voltage coefficient *g*_eff_ = *d*_eff_/(ε′ε_0_) VmN^−1^, which indicates the performance of a material as a piezoelectric generator. Crystals with high piezoelectric coefficients and low dielectric permittivity originate high values of *g*_eff_. For NNDM4NA@PLLA fiber mat at room temperature ε′ = 3.5 (Figure 5a), originating an effective piezoelectric voltage coefficient as high as *g*_eff_ = 4.1 VmN^−1^. This value is one order of magnitude higher than that obtained for the 2-dimensional hybrid ferroelectric perovskite (4-aminotetrahydropyran)2 lead tetrabromide, (ATHP)_2_PbBr_4_, which exhibited an already large value of *g*_eff_ = 0.67 VmN^−1^ [40] twice that displayed by a polyvinylidene fluoride (PVDF) polymer thin film for which *g*_eff_ = 0.29 VmN^−1^ [63]. It is also interesting to compare the above values with those of biological dipeptide derivatives of diphenylalanine electrospun fibers embedded in PLLA, as indicated in Table 2.

The extraordinary NNDM4NA nanocrystals’ piezoelectric response to applied periodical stress results from its high molecular dipole moment, low crystallographic symmetry (point group 2), high tensile strength and Young modulus, as well as its low dielectric permittivity, combined with the piezoelectric properties of the PLLA polymer matrix. In this work, we demonstrated that electrospun NNDM4NA@PLLA fibers are attractive composite nanomaterials as generators of high piezoelectric voltage as well as blue luminescence. Therefore, they may find applications in wearable devices or electronic textiles as two examples [67].

## 4. Conclusions

Highly aligned poly-l-lactic acid polymer microfibers with embedded *N*,*N*-dimethyl-4-nitroaniline acentric nanocrystals, fabricated using the electrospinning nanofabrication technique, were successfully produced. The microfibers are smooth, with a mean diameter of 1.38 µm, containing the nanocrystals spread evenly in their interior. The hybrid composite fibers display an extraordinarily high piezoelectric output response, where for a small stress of 5.0 × 10^3^ Nm^−2^, an effective piezoelectric voltage coefficient of *g*_eff_ = 4.1 VmN^−1^ is obtained. This results from the low room temperature dielectric permittivity of 3.5 combined with one of the highest effective piezoelectric coefficients measured on organic crystals formed by electron donor–acceptor molecules (*d*_eff_ = 220 pCN^−1^). Compared with other piezoactive compounds, such as poly(vinylidenefluoride), (PVDF) polymer (*g*_33_ = 0.29 VmN^−1^), ferroelectric molecular perovskite (4-aminotetrahydropyran)_2_ lead bromide ((ATHP)_2_PbBr_4_) (*g*_33_ = 0.67 VmN^−1^), the effective piezoelectric output response of poly-l-lactic acid microfibers with embedded *N*,*N*-dimethyl-4-nitroaniline is one order of magnitude higher. The fibers display solid-state blue fluorescence with a long lifetime decay of 147 ns behaving as luminescent nano-emitters. Furthermore, they exhibit an average increase of 67% on the Young modulus reaching 55 MPa, while the tensile strength reaches 2.8 MPa when compared to pure poly-L-lactic acid fibers, increasing the microfiber’s mechanical properties. The results show that organic nanocrystals, from small organic molecules, with elastic and piezoelectric properties form a hybrid functional 2-dimensional luminescent array, are mechanically strong and generate high output piezoelectric voltages, making them promising for applications in energy harvesting or as sensors through the piezoelectric effect, and as solid-state blue emitters.

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
