# Peer review of "High Piezoelectric Output Voltage from Blue Fluorescent N,N-Dimethyl-4-nitroaniline Nano Crystals in Poly-L-Lactic Acid Electrospun Fibers"

_materials, 2022, doi:10.3390/ma15227958_

Round 1

Reviewer 1 Report

I found this article to be very dense and difficulty to understand.  Since this journal has a broad scope and audience, I encourage the authors to make it more accessible.  For example, inclusion of molecular diagrams of the key components PLA and NNDM4NA (and non-piezo analog) emphasizing the conformations that contribute to performance would be very useful.  Additionally, comparison to other piezoelectric nanofibers would help support the claims of 'extraordinarily high piezoelectric output response' for those that don't have the numbers memorized.  Also, some comparisons or examples of uses would help to support the significance of blue fluorescence - is that useful and important or just a cool side property?  Are solid state blue emitters a thing that researchers are searching for?  

Author Response

We thank you very much for your helpful comments and suggestions, which we address below.

For example, inclusion of molecular diagrams of the key components PLA and NNDM4NA (and non-piezo analog) emphasizing the conformations that contribute to performance would be very useful

Author´s Response: The molecular structures have been included in the scheme 1 and referred to the scheme in the manuscript.

“Scheme 1. Molecular structures of (a) 4-nitroaniline (4NA), (b) 3-nitroaniline (3NA), (c) 2-methyl-4-nitroaniline (MNA), (d) N,N-dimethyl-4-nitroaniline (NNDM4NA) and (e) Poly (L-lactic acid) (PLLA).”

Additionally, comparison to other piezoelectric nanofibers would help support the claims of 'extraordinarily high piezoelectric output response' for those that don't have the numbers memorized. 

Author´s Response: The comparison with the other organic materials is made in Table 2, and throughout the text from lines 358 to 375.

Also, some comparisons or examples of uses would help to support the significance of blue fluorescence - is that useful and important or just a cool side property?  Are solid state blue emitters a thing that researchers are searching for?  

Author´s Response: A sentence explaining the importance of solid-state blue emission is added in the manuscript, lines 237 to 241.Three references are added.

“The emission at blue wavelengths of organic fluorophores in their condensed phase is of great scientific importance in materials science, towards their application as optical sensors and in luminescent and light-emitting devices [44-47]. Only a few compounds have a significant increase in light emission in their solid state, which overcomes the problem of aggregation quenching.”

Reviewer 2 Report

Reviewer
Comments and Suggestions for Authors
The manuscript is suitable for publication in your prestigious journal. However, there are some comments that should be considered such as:

1.    The authors should avoid writing the abbreviation in the abstract i-NNDM4NA.

2.    The authors should make a sequence in the entire manuscript such as somewhere fibers were written and somewhere microfibers.

3.    The authors should short the Keywords section such as composites, functional organic materials.

4.    It suggested that the abstract would be revised. In current form abstract is not readable.

5.    graphical abstract, must be added.

6.    English style and grammar should be carefully revised.

7.    Show the novelty in the introduction section.

8.    Authors should avoid writing the abbreviation in the materials and method section i-NNDM4NA.

9.    In SEM images reviewer can’t find the neat SEM images of microfibers.

10.  It suggested that the show the effect on diameter of NNDM4NA@PLLA piezoelectric load.

11. As shown in Figure 1. (a) Electrospun NNDM4NA@PLLA nanofiber. What does mean by nanofiber? Even the text authors discussed the results of microfibers.

12.  FTIR study added and discussed in manuscript.

13. It is suggested that TEM study must be added.

14.  In the entire manuscript authors make confusion about microfiber, nanofiber and fiber, kindly improve it.

Author Response

We thank you very much for your helpful comments and suggestions, which we address below.

  1. The authors should avoid writing the abbreviation in the abstract i-NNDM4NA.

Author’s response: The abbreviation NNDM4NA in the abstract is removed.

  1. The authors should make a sequence in the entire manuscript such as somewhere fibers were written and somewhere microfibers.

Author’s response: The word fibers has been changed to microfibers where appropriated.

  1. The authors should short the Keywords section such as composites, functional organic materials.

Author’s response: The word composites in keywords is removed.

  1. It suggested that the abstract would be revised. In current form abstract is not readable.

Author’s response: The abstract is revised and made more easy to read. It is now:

Abstract: N,N-dimethyl-4-nitroaniline is a superelastic and superplastic charge transfer molecular crystal that crystallizes in an acentric structure. Highly aligned poly-l-lactic acid polymer microfibers with embedded nanocrystals are fabricated using the electrospinning technique. The composite fibers display an extraordinarily high piezoelectric output response, where for a small stress of 5.0x103 Nm-2, an effective piezoelectric voltage coefficient of geff=3.6 VmN-1 is obtained. The fibers show solid-state blue fluorescence with a long lifetime decay (147 ns) lifetime decay. Furthermore, they exhibit an average increase of 67% in the Young modulus compared to polymer microfibers alone, reaching 55 MPa, while the tensile strength reaches 2.8 MPa. The results show that nanocrystals from small organic molecules, with elastic and piezoelectric properties, form a mechanically strong hybrid functional 2-dimensional array, promising for applications in energy harvesting and as solid-state blue emitters.”

  1. graphical abstract, must be added.

 Author’s response: A graphical abstract is included.

  1. English style and grammar should be carefully revised.

Author’s response: English and grammar are revised throughout the manuscript.

  1. Show the novelty in the introduction section.

Author’s response: A sentence is included in the introduction… “In this manuscript we explore the optical and piezoelectric properties of the N,N-dimethyl-4-nitroaniline nanocrystals as hybrid functional systems.”

  1. Authors should avoid writing the abbreviation in the materials and method section i-NNDM4NA.

Author’s response: The abbreviation has been substituted by IUPAC name, N,N-dimethyl-4-nitroaniline.

  1. In SEM images reviewer can’t find the neat SEM images of microfibers.

Author’s response: SEM images are now more clearly presented in: “Figure 1. (a) Electrospun NNDM4NA@PLLA nanofiber array as collected on a rotatory drum; (b) SEM images and diameter distributions with average fiber diameters (Mean), standard deviation (SD) and full width at half maximum (FWHM) at 250x (c) and 50,000x (d) magnification level. The fiber diameter distribution was fit to a log-normal function.”

  1. It suggested that the show the effect on diameter of NNDM4NA@PLLA piezoelectric load.

Author’s response: There is a spread in the microfiber diameters ranging from 0.8 to 2.0 µm, Figure 1b. In the piezoelectric measurements, the force is uniformly applied over a microfiber mat with 4 cm2 area, therefore encompassing a large number of microfibers.

  1. As shown in Figure 1. (a) Electrospun NNDM4NA@PLLA nanofiber. What does mean by nanofiber? Even the text authors discussed the results of microfibers.

Author’s response: The word nanofiber is replaced by microfibers throughout all the text.

  1. FTIR study added and discussed in manuscript.

Author’s response: The FTIR study is now integrated in main text, lines 253 to 279.

“3.3. Attenuated Total Reflectance Fourier Transform Infrared Spectroscopy (FTIR-ATR)

The Fourier transformed infrared (FTIR) analysis of electrospun fibers and NNDM4NA crystalline powder was recorded on a Jasco 4100 FTIR spectrometer in transmittance mode, in the range of 4500 to 600 cm-1, by averaging 32 scans and using a resolution of 8 cm-1. FTIR data were treated with OriginPro 2017 SR2 software (OriginLab Corporation, USA). The incorporation of NNDM4NA in PPLA polymeric fibers was confirmed spectroscopically by the collected data, as indicated in Figure 3. NNDM4NA exhibits the characteristics of conjugated double bonds with typical bands within the ranges 1660-1480 cm-1, which is consistent with a high degree of push-pull structural character (donor-acceptor). NNDM4NA, as a tertiary amine, has C-N stretching bands from 1250 cm-1 to 1020 cm-1. In the NNDM4NA spectrum, two bands appear at 1060 cm-1 and 1095 cm-1, with weak to medium intensity, due to the lack of polarity of the C-N bond. These bands overlap with the more intense PLLA bands in the NNDM4NA@PLLA spectrum. For aromatic nitro compounds, the asymmetric and symmetric stretching vibration of the NO2 group are assigned to an intense band in the region around 1590–1600 cm-1 and 1380-1390 cm-1, which is due to the asymmetric and symmetric stretching vibration of the group [48-52]. The symmetric stretching vibration at 1304 cm-1 of NNDM4NA is shifted to 1315 cm-1 in NNDM4NA@PLLA [53-54]. The bands in the 1123 cm-1 and 1177 cm-1 region are assigned to bending vibrations of the C-H and C-C. Weak bands at 1432 cm-1 and 1462 cm-1 in NNDM4NA are not present in NNDM4NA@PLLA, due to overlapping with the intense PLLA band at 1452 cm-1. The band at 1502 cm-1 in NNDM4NA, also associated with NO2 asymmetric stretching vi-bration, is shifted to 1529 cm-1 in NNDM4NA@PLLA. The weak 1540 cm-1 band in NNDM4NA is hidden by the polymer matrix in NNDM4NA@PLLA. In the NNDM4NA spectrum, the intense band at 1657 cm-1, which is mainly due to the in-plane bending modes of N(CH3)2, is absent in the NNDM4NA@PLLA spectrum. The reason must be that the nanocrystals are intercalated in the polymer chains and those groups are con-strained inside the matrix.”

  1. It is suggested that TEM study must be added.

Author’s response: The manuscript is already very long and it is clearly shown in the XRD graphics, that organic nanocrystals are crystalized and fully embedded in polymer matrix. Therefore, we are not performing TEM measurements.

  1. In the entire manuscript authors make confusion about microfiber, nanofiber and fiber, kindly improve it.

Author’s response: The manuscript was cheeked for microfiber, nanofiber and fiber and changed uniformly to microfibers.

Round 2

Reviewer 1 Report

The paper appears to be significantly improved.  I'm glad the reviews were useful in that way.

In comparing the piezoelectric performance, the authors only include very similar materials.  Please also include PVDF nanofibers and liquid crystal nanofibers in the comparison table.  These have been studied extensively and are relevant for comparison in both the piezoelectric and sensing aspects of this paper - particularly given the claims of outstanding performance of the described material.

Author Response

Response to Reviewer 1

The authors thank you very much for your helpful comments and suggestions, which we address below.

Question: In comparing the piezoelectric performance, the authors only include very similar materials.  Please also include PVDF nanofibers and liquid crystal nanofibers in the comparison table.  These have been studied extensively and are relevant for comparison in both the piezoelectric and sensing aspects of this paper - particularly given the claims of outstanding performance of the described material.

Authors response: We modified Table 2, including PVDF polymer thin film, a perovskite and an amino-acid thin film. We are not aware of liquid crystals uses as nanogenerators through the piezoelectric effect. Liquid crystals have however been demonstrated to perform as triboelectric energy generators, and therefore we think cannot be compared as the piezoelectric and triboelectric effects are not the same (see for example 10.1021/acsaelm.1c01307; 10.1016/j.scib.2021.05.016; 10.1002/adfm.201808633)

Table 2.

Active system

Force/area [Nm2]

Vout [V]

deff [pCN-1]

geff [VmN-1]

Ref.

NNDM4NA

(Fiber array)

4x103

33

147

4.1

This work

3NA

(Fiber array)

4x103

7

37

1.2

[34]

(ATHP)2PbBr4

(Thin film)

__

__

__

0.67

[40]

PVDF

(Thin film)

__

__

__

0.29 (g33)

[63]

PVDF

(Fiber array)

­­__

__

25 (d33)

__

[64]

L-Tyrosine Film

86 x104

0.5

__

__

[65]

Boc-PhePhe

(Fiber array)

4x103

30

8.4

0.3

[32]

Boc-pNPhepNPhe (Fiber array)

4x103

58

16

0.6

[66]

Reviewer 2 Report

The authors well executed the revision but the manuscript needs substantial writing improvements.

1- The abstract should be revised, please include the main problem and why this research was carried out, and synchronize the flow of text with results directing to the potential for application authors mentioned in this article.

2- Please relate current research with some latest references in the introduction section and delete all the poor references such as Mater. Sci. 1972, 7, 31-33, Philos. Mag. B 1992, 66, 293-305, Chem. Phys. 1992, 97, 5616-5630, Phys. Rev. B 1991, 43, 14683-14691, Appl. Phys. 1979, 50, 2523-2527, Opt. Commun. 1975, 15, 258-262, Acta Crystallogr. 1965, 18, Mater. Sci. 1972, 7, 31-33 and so on...

3- Please include suitable references next to claiming and reasoning statements.4- Please carefully check and correct typos/grammatical errors.5- The quality graphical abstract is poor, please revise.

6- The conclusion is vague, please revise it with significant value.

Author Response

Response to Reviewer 2

The authors thank the reviewer for the comments and valuable suggestions. The following is a point-to-point response.

Point 1- The abstract should be revised, please include the main problem and why this research was carried out, and synchronize the flow of text with results directing to the potential for application authors mentioned in this article.

Authors response: the Abstract is modified following the reviewer sugestions. It reads now:

Abstract: N,N-dimethyl-4-nitroaniline is a piezoelectric organic superplastic and superelastic charge transfer molecular crystal that crystallizes in an acentric structure. Organic mechanical flexible crystals are of great importance as they stand between soft matter and inorganic crystals. Highly aligned poly-l-lactic acid polymer microfibers with embedded N,N-dimethyl-4-nitroaniline nanocrystals are fabricated using the electrospinning technique and its piezoelectric and optical properties are explored as hybrid systems. The composite fibers display an extraordinarily high piezoelectric output response, where for a small stress of 5.0x103 Nm-2, an effective piezoelectric voltage coefficient of geff=4.1 VmN-1 is obtained which is one of the highest among piezoelectric polymers and organic lead perovskites. Mechanically, they exhibit an average increase of 67% in the Young modulus compared to polymer microfibers alone, reaching 55 MPa, while the tensile strength reaches 2.8 MPa. Furthermore, the fibers show solid-state blue fluorescence, important for emission applications, with a long lifetime decay (147 ns) lifetime decay. The present results show that nanocrystals from small organic molecules with luminescent, elastic and piezoelectric properties form a mechanically strong hybrid functional 2-dimensional array, promising for applications in energy harvesting through the piezoelectric effect and as solid-state blue emitters.”

Point 2- Please relate current research with some latest references in the introduction section and delete all the poor references such as Mater. Sci. 1972, 7, 31-33, Philos. Mag. B 1992, 66, 293-305, Chem. Phys. 1992, 97, 5616-5630, Phys. Rev. B 1991, 43, 14683-14691, Appl. Phys. 1979, 50, 2523-2527, Opt. Commun. 1975, 15, 258-262, Acta Crystallogr. 1965, 18, 

Authors response: We would like to call the reviewer attention that references 1, 8, 10, 11, 12 and 15 are of fundamental importance, as they demonstrated and established the crystallographic, linear and nonlinear optical as well piezoelectric properties of small organic charge-transfer molecular crystals. We therefore consider that, although old, they are still actual and deserve to be referenced.

The previous reference 13 (Oudar, J. L.; Le Person, H. Second-order polarizabilities of some aromatic molecules. Opt. Commun. 1975, 15, 258-262.) has now been replaced by a more recent reference: “Dmitriev, V. G.; Gurzadyan, G. G.; Nikogosyan, D. N., Handbook of nonlinear optical crystals. Springer: 2013; Vol. 64.”

Point 3 - Please include suitable references next to claiming and reasoning statements.

Authors response: We think that we have referenced appropriately the manuscript claims which in our opinion has already a significant number of references. Those references are: [44], [33], [45-48], [49-53], [54-55], [17], [56], [57-58], [34], [61-62], [40], [63], [64], [65], [66], [67].

Point 4 - Please carefully check and correct typos/grammatical errors.

Authors response: The manuscript was checked again for grammatical errors.

Point 5 - The quality graphical abstract is poor, please revise.

Authors response: The quality of graphical abstract quality is improved. It is now:

Point 6- The conclusion is vague, please revise it with significant value.

Authors response: The conclusions are revised as suggested. It reads now:

Highly aligned poly-l-lactic acid polymer microfibers with embedded N,N-dimethyl-4-nitroaniline acentric nanocrystals, fabricated using the electrospinning nanofabrication technique, are successfully produced.  The microfibers are smooth, with a mean diameter of 1.38 µm, containing the nanocrystals evenly spread in their interior. The hybrid composite fibers display an extraordinarily high piezoelectric output response, where for a small stress of 5.0x103 Nm-2, an effective piezoelectric voltage coefficient of geff = 4.1 VmN-1 is obtained. This results from the low room temperature dielectric permittivity of 3.5 combined with one of the highest effective piezoelectric coefficients measured on organic crystals formed by electron donor-acceptor molecules (deff = 220 pCN-1).  Compared with other piezoactive compounds, such as poly(vinylidenefluoride), (PVDF) polymer (g33 = 0.29 VmN-1), ferroelectric molecular perovskite    di-(4-aminotetrahydropyran) lead bromide ((ATHP)2PbBr4) (g33 = 0.67 VmN-1), the effective piezoelectric output response of poly-l-lactic acid microfibers with embedded N,N-dimethyl-4-nitroaniline is one order of magnitude higher. The fibers display solid-state blue fluorescence with a long lifetime decay of 147 ns behaving as luminescent nano-emitters. Furthermore, they exhibit an average increase of 67% on the Young modulus reaching 55 MPa, while the tensile strength reaches 2.8 MPa when compared to pure poly-L-lactic acid fibers, increasing the microfibers mechanical properties. The results show that organic nanocrystals, from small organic molecules, with elastic and piezoelectric properties form a hybrid functional 2-dimensional luminescent array, which are mechanically strong and generate high output piezoelectric voltages, making them promising for applications in energy harvesting or as sensors, through the piezoelectric effect, and as solid-state blue emitters.
